# Contribution of UltraFast™ Ultrasound and Shear Wave Elastography in the Imaging of Carotid Artery Disease

**DOI:** 10.3390/diagnostics12051168

**Published:** 2022-05-08

**Authors:** Antonio Bulum, Gordana Ivanac, Filip Mandurić, Luka Pfeifer, Marta Bulum, Eugen Divjak, Stipe Radoš, Boris Brkljačić

**Affiliations:** 1Department of Diagnostic and Interventional Radiology, University Hospital Dubrava, Avenija Gojka Šuška 6, 10000 Zagreb, Croatia; gordana.augustan@gmail.com (G.I.); luka.pfeifer.os@gmail.com (L.P.); mchubela@gmail.com (M.B.); edivjak@gmail.com (E.D.); rados09@gmail.com (S.R.); boris@brkljacic.com (B.B.); 2School of Medicine, University of Zagreb, Šalata 3, 10000 Zagreb, Croatia; 3Institute of Emergency Medicine of the City of Zagreb, Heinzelova 88, 10000 Zagreb, Croatia; filip.manduric5@gmail.com

**Keywords:** carotid artery, carotid plaque, ultrasound, ultrafast, elastography, imaging

## Abstract

Carotid artery disease is one of the main global causes of disability and premature mortality in the spectrum of cardiovascular diseases. One of its main consequences, stroke, is the second biggest global contributor to disability and burden via Disability Adjusted Life Years after ischemic heart disease. In the last decades, B-mode and Doppler-based ultrasound imaging techniques have become an indispensable part of modern medical imaging of carotid artery disease. However, they have limited abilities in carotid artery plaque and wall characterization and are unable to provide simultaneous quantitative and qualitative flow information while the images are burdened by low framerates. UltraFast™ ultrasound is able to overcome these obstacles by providing simultaneous quantitative and qualitative flow analysis information in high frame rates via UltraFast™ Doppler. Another newly developed ultrasound technique, shear wave elastography, is based on the visualization of induced shear waves and the measurement of the shear wave propagation speed in the examined tissues which enables real-time carotid plaque and wall analysis. These newly developed ultrasound modalities have potential to significantly improve workflow efficiency and are able to provide a plethora of additional imaging information of carotid artery disease in comparison to conventional ultrasound techniques.

## 1. Carotid Pathology

Cardiovascular diseases are globally a main cause of disability and premature mortality, with atherosclerosis as the main pathological process. Atherosclerosis begins early during one’s lifetime and predominantly remains asymptomatic until it is already in an advanced stage [1]. This is especially apparent in carotid artery disease, where advanced disease may be latent until the onset of life-threatening conditions, such as stroke, the second largest cause of death globally after ischemic heart disease [2]. In addition to being the second leading cause of mortality, nonfatal episodes of stroke are a major global contributor to disability, and stroke burden via DALY (Disability Adjusted Life Years) is continuously increasing. For 2019, the data show that ischemic and hemorrhagic stroke were the second and third leading cause of DALY, respectively [3].

As mentioned previously, stroke can be divided into ischemic stroke and hemorrhagic stroke, which is further divided into intracerebral and subarachnoid hemorrhage. Ischemic stroke is defined as infarction of the brain, spinal cord or retina and encompasses around 70% of all strokes globally. The majority of them are thromboembolic in origin, with the most common sources of embolism being large artery atherosclerosis and cardiac diseases [4]. Less common causes include arterial dissection, small vessel disease and inflammatory diseases, such as vasculitis. In patients with large artery atherosclerosis, the embolism typically originates from atherosclerotic changes om the internal carotid artery immediately after its origin from the bifurcation of the common carotid artery.

The main risk factors for the development of carotid atherosclerotic changes are the same ones responsible for the development of cardiovascular diseases in general. These include dyslipidemia, hypertension, obesity, smoking and sedentary lifestyle, as well as advanced glycation end products (AGEs) and its receptors and C-reactive protein (CRP). AGEs are endogenously or exogenously formed glycation products of a protein or lipoprotein. Their blood levels increase with age, with smoking accelerating that process, and they serve as biomarkers of severity of the cardiovascular disease, both carotid and coronary [5]. On the other hand, CRP is an acute phase protein, whose blood levels rise in inflammatory conditions such as infections but also in non-infectious systemic diseases, such as atherosclerosis. The more blood vessels are affected, the higher CRP levels can be detected [6]. 

The three main pathological processes causing carotid atherosclerotic changes are: increased carotid intima-media thickness, carotid plaque formation and carotid stenosis. In 2020, the most common of these pathologies was increased carotid intima-media thickness, whose global prevalence in the age groups of 30–79 years is 27.6% [1].

Carotid artery wall intima-media-thickness (IMT) is a surrogate measure of carotid atherosclerosis. IMT is measured non-invasively, by ultrasound, as the distance between the lumen–intima and media–adventitia interfaces of the arterial wall. The IMT of the common carotid artery (CCA) and the proximal portion of the internal carotid artery (ICA) can both be measured, although, in principle, the measurement of the IMT of the CCA is considered simpler and more appropriate for assessing overall cardiovascular risk [7]. The recent American College of Cardiology Foundation–American Heart Association guidelines gave carotid intima–media thickness a level IIa recommendation for cardiovascular risk evaluation, with an indication of high risk if the common carotid artery IMT is above the 75th percentile [8].

Carotid artery stenosis is defined as a narrowing of its lumen and may clinically be symptomatic or asymptomatic. Symptomatic stenosis is experienced by patients who have suffered a transient ischemic attack (TIA) or stroke due to lumen narrowing or thromboembolic events from the affected artery. It is considered that carotid artery stenosis is responsible for about 10–20% of all strokes, and there has been a long-standing consensus regarding their management [2]. The North American Symptomatic Carotid Endarterectomy Trial (NASCET) and the European Carotid Surgery Trial (ECST), both published in 1991, outlined the importance of surgery in different groups of symptomatic patients [9].

The diagnosis of asymptomatic carotid stenosis is made when a lumen narrowing is found without the evidence of previous TIA, stroke or focal neurological in the course of 6 months ago [10]. There is still no true consensus regarding the best treatment of asymptomatic carotid artery stenosis, and the latest European Society for Vascular Surgery (ESVS) guidelines recommend that patients with an average surgical risk and an asymptomatic carotid artery stenosis of 60–99% should be considered for carotid endarterectomy (CEA) only in the presence of one or more characteristics that may be associated with an increased risk of late ipsilateral stroke. Carotid artery stenting (CAS) is also feasible in these patients; however, it is indicated over CEA for patients with a high risk of surgery retaining the increased risk of late ipsilateral stroke [11].

Carotid plaque is a focal structure that affects the lumen of an artery (protrudes into it) for a minimum of 5 mm or 50% of the surrounding IMT or demonstrate a thickness greater than 1.5 mm as measured from the media–adventitia interface to the intima–lumen interface. In comparison to IMT, a carotid plaque is a different atherosclerotic process indicating increased vascular risk over IMT, occurring most commonly at the carotid bifurcation, and is usually indirectly identified during measurements of luminal stenosis [12,13]. Atherosclerotic plaques most frequently occur in the areas of arterial bifurcations, branching regions and pronounced convolutions of blood vessels. In addition, areas of bifurcation are sites where increased intima–media thickness is often encountered [14]. Owing to advances in vascular imaging, many additional features of carotid plaques can nowadays be identified, including calcifications, fibrous caps, intraplaque hemorrhage and others. This has made a shift in the process of diagnosing carotid atherosclerotic disease from only quantifying the degree of stenosis to plaque characterization, which allows better patient risk stratification and can guide future management [15].

To understand plaque formation, it is important to define laminar and disturbed laminar flow. During the laminar flow, fluid (e.g., blood) moves in concentric laminae that are parallel to the course of the blood vessel, and such flow is typical in the straight parts of arteries. Here, the fluid flow velocity is highest at the center and falls towards the periphery of the vessel. Disturbed laminar flow is characterized by streamline deviations, recirculation, turbulence and reattachment to the forward flow. Such flow typically occurs at branching sites, bifurcations and geometric wall irregularities [16].

Due to disturbances of laminar blood flow and turbulences, as well as vessel irregularities, atherosclerotic plaques in the carotid system mainly occur in the area of bifurcation of the CCA to the external carotid artery (ECA) and ICA. The ostium of the ICA is also frequently affected, primarily the posterior wall of the carotid sinus while the intracranial branches are affected less frequently, especially in Caucasian populations. However, intracranial arterial atherosclerosis and its consequences are probably the most common stroke subtype worldwide due to its high prevalence in the Asian population, which constitutes the majority of the world’s population [17].

## 2. UltraFast™ Ultrasound

UltraFast™ (SuperSonic Imagine, Aix en Provence, France) imaging is the new step into the development of medical ultrasonography that allows not only 100 times higher frame rates than conventional ultrasound scanners but also scanning the whole region of interest in one single insonification. In contrast, current ultrasound systems are based on a serialized architecture, and imaging is performed using a series of insonifications of focused beams, and each beam is then used to reconstruct one plane of an image, meaning that the frame rate is limited by the time it takes for echoes to be transmitted, received and processed for all lines of the image. 

UltraFast™ approaches this problem with a concept of computing in parallel as many planes as possible so it is able to compute a full image from a single transmit, regardless of the image size and its other characteristics. By using a parallel architecture, the frame rate is no longer limited by the number of planes reconstructed but by the time an ultrasound pulse needs to travel through the insonated tissues and back to the probe [18].

There are several ways in which an ultrafast imaging architecture is conceptualized: the one used for UltraFastTM is based on the use of plane wave insonification [19].

By applying flat delays on the transit elements of an ultrasound probe, a plane wave is created, after which the generated wave insonifies the entire area of interest (ROI) at once. The echoes sent back to the probe are then registered and processed by the computer in the ultrafast scanner, creating an image of the examined area, all in one transit.

The idea of ultrafast ultrasound was first introduced more than 40 years ago but was limited by the processing technology at the time. Thanks to the advancements of the processing power of personal and commercial computers in the last decades, it is recently being implemented more often in commercial ultrasound diagnostic devices. As mentioned previously, it has bypassed the low frame rate limitation of conventional ultrasound imaging by significantly reducing the number of insonifications required to generate an equivalent image. This has allowed for the development of a series of new ultrasound imaging modalities, such as shear wave elastography (SWE), ultrafast Doppler and ultrafast pulse wave velocity (ufPWV) [20].

Linear transducers used for UltraFast™ imaging use plane waves that propagate through the whole region of interest, meaning that one plane wave is enough to cover the whole region of interest. Tissue contrast and resolution can be improved by sending several tilted plane waves, backscattering the echoes and summing and processing them by the scanner. In other words, the more waves there are, the higher quality of image is achieved. It has been shown that just nine different angles of plane waves transmitted by the transducer are enough to make an image with equal resolution and sensitivity compared to the conventional ultrasound, meaning that less time is required to perform an equally adequate examination [21].

## 3. Shear Wave Elastography

In addition to conventional, Doppler and contrast-enhanced ultrasonography, elastography techniques are another area where ultrasound is being increasingly used, this time to assess the elastic properties of the insonated tissue. There are two types of ultrasound elastography: strain and shear wave elastography (SWE). Strain elastography is performed by manual compression using the transducer, which then produces an image based on the resulting displacement of the tissue caused by the compression. However, it is difficult to measure the exact amount of the applied force during compression, resulting in the method being difficult to standardize. Additionally, the absolute elasticity values cannot be calculated, and only qualitative results can be obtained. Unlike strain elastography, SWE is a type of ultrasound elastography where the elastic properties of the insonated tissues can be expressed both qualitatively and quantitatively (Figure 1 and Figure 2) [22].

In order to understand the principles of SWE elastography, we have to remember that as the ultrasound pulse passes through tissues, it can be reflected, absorbed or it can create tissue displacement. The latter induces mechanical vibrations resulting in shear waves, which are perpendicular to the primary ultrasound wave and propagate through the tissues with measurable speed. Since the velocity of the shear waves is proportional to tissue elasticity (shear waves propagate faster in stiff tissue), quantitative values of tissue stiffness can be obtained and expressed by the Young’s modulus, the imaging mode being called SWE. SWE is based on two concepts, acoustic radiation force generating shear waves, whose velocity is linked to the tissue stiffness, and conventional ultrasound imaging that allows the necessary temporal resolution to film shear wave propagation. If it used in combination with UltraFast™ imaging, tissue stiffness can be quantified and displayed in real-time in addition to B mode ultrasound [23].

This technique allows the characterization of carotid plaque composition and stability. For example, areas with higher stiffness correspond to calcifications within the plaque, and areas with lower stiffness values to lipid deposits or intraplaque hemorrhage, both indicating characteristics of vulnerable plaques [24]. In addition to that, stiffness values were correlated with the Gray–Weale classification of plaques based on echogenicity and were proven to be statistically significantly lower in symptomatic compared to asymptomatic patients [25]. SWE has also already proven its clinical correlation with stiffness values of the carotid artery wall measured using SWE having a positive correlation with known risk factors of atherosclerosis [26]. A systematic review comparing 19 studies concluded that there is a great potential to improve stroke risk stratification using SWE as a tool for the assessment of arterial walls, plaque stiffness and plaque vulnerability. All studies demonstrated the ability of SWE to quantitatively assess the stiffness of the arterial walls and plaques and to assess plaque vulnerability based on echogenicity, symptomatology and histology while also having good to excellent reproducibility [27].

In addition to the assessment of carotid arteries, SWE found its use in numerous branches of medicine, from hematology to breast imaging and thyroid mass identification, with promising results in gynecological, urological, neurological and dermatological uses [28,29,30,31].

## 4. UltraFast™ Doppler

One of the main abilities of ultrasound as a medical imaging modality is tracking blood flow dynamics in real time. Because of that Doppler based techniques of imaging are a mainstay on every modern medical ultrasound device, especially with the framework of cardiovascular imaging, which includes imaging of carotid artery disease. Conventional Doppler ultrasound imaging consists of two modes: Color flow imaging (CFI) and the Pulsed-wave Doppler (PW Doppler).

Tracking blood flow dynamics in CFI is achieved by estimating flow velocity. The flow velocity estimations are based on the use of a number of narrowband pulses which are transmitted at a constant pulse repetition frequency (PRF) in order to estimate the Doppler frequency. The main challenge of this method is how to separate the Doppler flow signals arriving at the probe from all the tissue echoes arriving at the same time and estimate the mean flow velocity. This was done mostly using correlation methods [32].

However, in order to quantify the Doppler signals acquainted via CFI PW Doppler is used. The quantification is achieved by performing an FFT type of spectral analysis which is used to deduce the distribution of Doppler velocities within a sample volume. The results are the displayed as a graph, with velocity as the y-axis and time at the x-axis [33].

To acquire data from these modes, the practitioner switches between them analyzing the PW Doppler at the locations pinpointed by the color Doppler image. With the introduction of UltraFast™ Doppler (SuperSonic Imagine, Aix en Provence, France), quantitative data can be assessed in all pixels at the same time. Because it relies on plane waves and not focused beams in the image acquisition, there is no time lag at the sides of the image. On a conventional ultrasound device, such quantitative analysis is only possible by limiting the region of interest (ROI) to a single acoustic line. UltraFast™ Doppler allows the merging of CFI and PW Doppler mode in a single acquisition (Figure 3).

High frame rates of UltraFast™ Doppler provide a high temporal resolution on CFI, which enables the visualization of more complex, as well as fast flows. Such flow patterns can be seen in more detail, consequently enabling the examiner to establish more accurate diagnosis. To better visualize the slow flows, high frame rates are traded for higher sensitivity and better spatial resolution. This way some small vessels can only be seen using UltraFast™ Doppler and without the need for intravenous contrast agents. 

The high temporal resolution enables a consistent flow of information in the entire imaged area since the received Doppler signals of all the pixels in the image are acquired at the same time. In contrast, conventional color Doppler images are sequentially acquired, the Doppler signals on the edges of the color box arrive to the probe with a delay up to several hundreds of milliseconds in comparison to UltraFast™ Doppler [18].

When performing an UltraFast Doppler examination, a single-shot acquisition mode is usually initiated from the conventional color Doppler imaging mode. After that, a range of UltraFast Doppler data is acquired (typically 2 to 4 s), and the picture is frozen. The examiner can then review the UltraFast color flow data, review a single or multiple frames offering the best image visualization of the flow characteristics of interest and simultaneously perform a retrospective spectral analysis of the color box. Furthermore, using UltraFast™ Doppler, a short clip of not only one, but multiple regions of interest can be obtained, providing a more precise comparison of both mean and peak flow velocity originating from the same cardiac cycle (Figure 4) [21].

A case report was published, which showed carotid artery plaque ulceration and intraplaque hemorrhage that could only be seen using the UltraFast™ Doppler, whereas the conventional Doppler and CT proved ineffective in detecting plaque changes [24].

Another area in which UltraFast™ Doppler might help guide the management of carotid artery disease by identifying carotid plaques exhibiting vulnerability is through the use of vector Doppler imaging. Unlike conventional ultrasound, which cannot accurately evaluate the flow speeds in contact with the arterial wall, UltraFast™ Doppler is able, through the use of vector Doppler imaging, to accurately measure flow velocity at any point in the image without the need for angle correction [34].

With its ability to evaluate the flow velocity at any point in the image, UltraFast™ Doppler can be used for the assessment of wall shear stress (WSS). WSS is defined as the tangential stress on the endothelial surface of the arterial wall derived from the friction of the flowing blood. It is directly proportional to the velocity of the blood flow and inversely proportional to the cube of the arterial radius and is thought to determine the site-specific predilection of atherosclerosis [16]. In addition to that, it has already been associated with carotid plaque inflammation and rupture [35].

Using UltraFast™ vector Doppler the WSS along a carotid atherosclerotic stenosis can be directly evaluated in a single cardiac cycle and in the future might be used to guide patient management, especially those with asymptomatic carotid artery plaques [36].

## 5. UltraFast™ Pulse Wave Velocity (PWV)

After the contraction of the ventricle and the opening of the aortic valve, a pulse wave is generated, which then passes through the aorta and carotid arteries, causing the arterial walls to dilate. The speed of the pulse wave can be measured and is called pulse wave velocity (PWV), which depends on artery wall thickness and elasticity. By being dependent on arterial wall thickness and elasticity, it has been shown that PWV correlates well with atherosclerotic risk and cardiovascular mortality [37]. Traditionally, brachial-ankle (baPWV) and carotid-femoral pulse wave velocity (cfPWV) have been advised as a non-invasive measure of arterial stiffness. However, these methods are time-consuming, require complex calculations and errors are easy to make because they require measurements of the pulse wave transit time between the carotid and femoral artery, resulting in their limited clinical use [38,39].

Unlike traditional baPWV and cfPWV, high imaging frame rates of UltraFast™ ultrasound enable the determination of local arterial stiffness in one cardiac cycle because pulse waves can be tracked in real-time [38]. This implies that imaging of the carotid arteries is sufficient for the assessment of atherosclerotic changes and there is no need to image other arteries, as in the measurement of carotid-femoral pulse wave velocity. It is non-invasive, and the acquisition can be performed in less than a minute, leaving no place for errors because of distance between the two separate imaging spots. 

In a single cardiac cycle, the estimation of velocity shows two accelerations peaks which correspond to two propagating waves. The first wave after the aortic valve opening at the beginning of the systole and the second wave after the closure of the aortic valve at the end of the systole. Therefore, PWV can be measured both at the beginning (PWV-BS) and the end of systole (PWV-ES) [37]. As suggested, PWV-ES is a better predictor of arterial stiffness than PWV-BS. It can be more accurately estimated than PWV-BS, and it represents the arterial stiffness at the systolic phase, which is more sensitive to changes in the arterial wall stiffness caused by age and other relative diseases [38].

Studies reported that PWV at the end of systole correlates with carotid IMT in arteries with early stages of atherosclerosis [37,38]. Arterial stiffness is positively related to age, blood pressure and hyperlipidemia [38,39]. One study compared IMT and PWV in patients with coronary artery disease (CAD) and concluded that UltraFast™ PWV is more sensitive in identifying patients with CAD and is of more clinical value [39]. Because there is no constant correlation between the increase in IMT and arterial stiffness, it is advised that both PWV and IMT should be conducted to establish the adequate diagnosis [38].

In contrast, a study comparing small numbers of normotensive and hypertensive patients found no correlation between PWV and patients’ age and blood pressure [26]. Furthermore, another study suggested that the analysis of carotid arteries with atherosclerotic plaque is of limited success, noting that the size of plaque may interfere with the propagation of pulse wave and that, in case of tortuous arteries, it may be hard to place the transducer parallel to the vessels, with difficulties in the precise measurement [18]. It is important to note that PWV-BS and PWV-ES values cannot be compared with values of PWV obtained by conventional imaging, because the latter are mean values of the artery of interest. In general, it is considered that the PWV values obtained by UltraFast™ ultrasound are lower than those yielded by conventional technologies. Additionally, the reflection wave originating from the intracranial arteries to the heart might influence the measured PWV value, and it is suggested to choose a straight arterial segment of the CCA that is proximal enough from the bifurcation of the common carotid artery in order to avoid this. The examiner should also have in mind that PWV by UltraFast™ ultrasound cannot be measured over an artery segment shorter than 50 mm. Therefore, multicenter studies with a large sample size and comparative studies comparing the measurement of PWV using UltraFast™ ultrasound with other established imaging modalities are needed to confirm its clinical usefulness [40].

## 6. Conclusions

In the imaging of carotid artery disease, B-mode and Doppler-based ultrasound techniques are well known and scientifically proven modalities of modern ultrasound imaging devices. They are used for the analysis of the carotid artery wall and blood flow, including its quantification, and are indispensable in the everyday workflow of contemporary medical imaging. While color Doppler provides a qualitative analysis of the mean flow velocity, PW Doppler provides quantification of the flow characteristics. However, they both have their disadvantages, with color Doppler sacrificing quantitative analysis in order to provide qualitative information over a large area of interest, while PW Doppler can provide flow quantification typically at only a single location at a time. In addition to that, the examiner has to constantly switch between methods, deal with flow imaging information over a large area of interest and flow quantification at only one location while the images are displayed in low framerates.

On the other hand, UltraFast™ ultrasound is able to overcome all these obstacles by merging color Doppler and PW Doppler into a single modality: UltraFast™ Doppler. The high frame rates provided by UltraFast™ Doppler have enabled new opportunities in vascular ultrasound imaging, including improved spatial and temporal resolution. UltraFast™ also enabled the visualization of induced shear waves and the measurement of the shear wave propagation speed, leading to the development of SWE, which enabled real-time carotid plaque and wall stiffness analysis. UltraFast™ imaging provides a great potential for clinical use in the diagnosis of carotid artery atherosclerosis, better understanding of plaque calcifications, and improved assessment of plaque rupture possibility.

UltraFast™ Doppler outperforms the visualization of complex blood flows, whether those are fast or slow flows, compared to the conventional Doppler. Furthermore, quantitative data of carotid blood flow are gathered in one cardiac cycle, and each pixel of the image is synchronous, with no time lag between them. Diagnosis can be made in less time with more accuracy, and intraplaque hemorrhage or slow blood flow into the walls of carotid arteries can be observed. The determination of PWV in carotid arteries using UltraFast™ Doppler is performed in less time and with fewer errors compared to conventional ultrasound. 

Cardiovascular diseases present a main cause of global disability and premature mortality, with carotid artery disease and its consequences at the forefront of the disease spectrum. Having taken this into consideration, UltraFast™ ultrasound has potential to have a major impact in the future of biomedical ultrasound and imaging. However, more studies and continuous clinical practice are necessary in order for UltraFast™ ultrasound to achieve its potential in making the initial diagnosis of carotid artery disease faster and more reliable. 

## Figures and Tables

**Figure 1 diagnostics-12-01168-f001:**
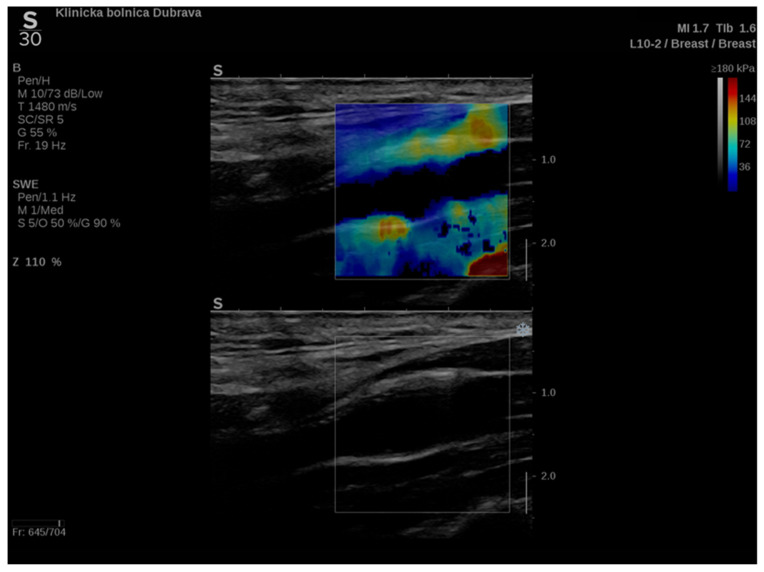
An example of an ultrasound examination with shear wave elastography of a segment of the distal right CCA in the longitudinal view with B-mode ultrasound at bottom and shear wave elastography at the top where the elastic properties of the examined tissues (carotid artery wall and surrounding soft tissues) are displayed qualitatively by benign color-coded and superimposed on the B-mode image. Red color denotes the stiffest areas with the highest elastic modulus values.

**Figure 2 diagnostics-12-01168-f002:**
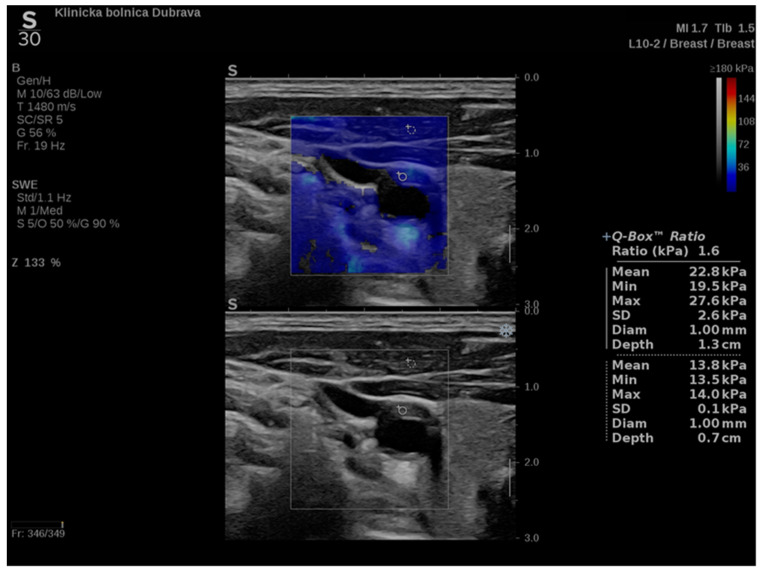
Another example of an ultrasound examination with shear wave elastography of a segment of the CCA, this time in the transverse view. A carotid artery plaque is visible in the wall of the CCA and displayed with B-mode ultrasound at the bottom and shear wave elastography at the top. In this figure, the elastic properties of the examined tissues are displayed both qualitatively and quantitatively. Quantitative measures are visible on the right-hand side of the picture, measured using two regions of interest, one centered over the plaque and another over the adjoining soft tissues, and an elasticity ratio between the two is calculated by the ultrasound device.

**Figure 3 diagnostics-12-01168-f003:**
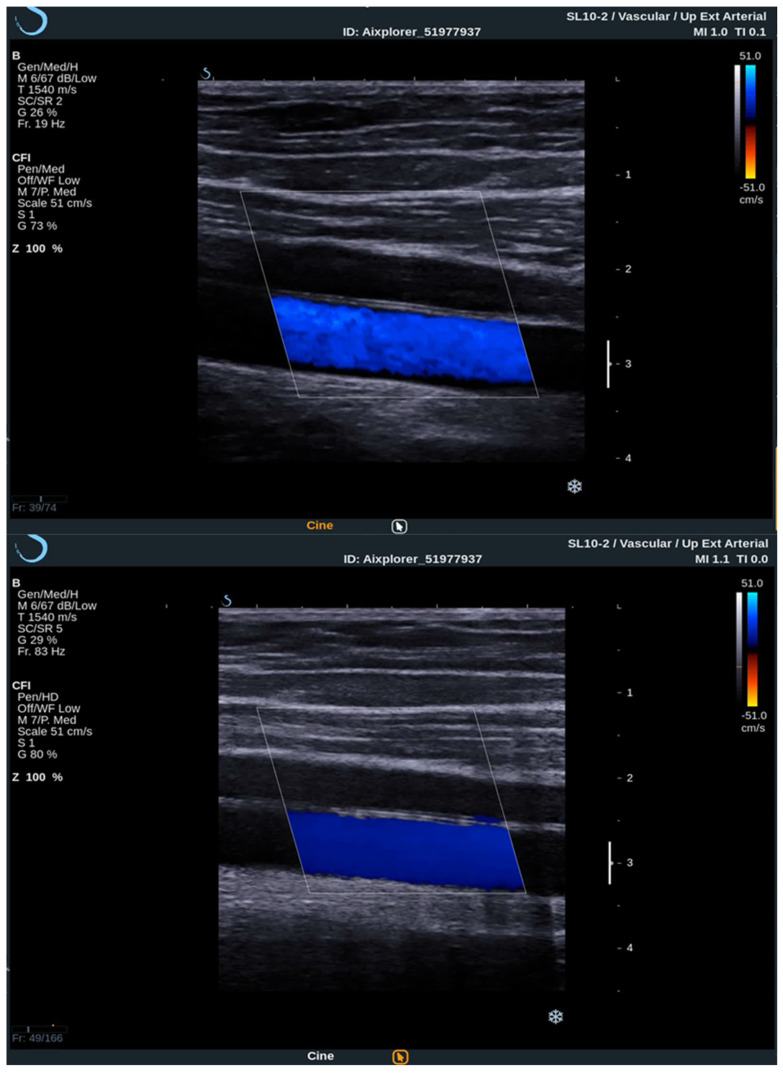
Comparison between conventional CFI Doppler ultrasound at the top and UltraFast™ Doppler ultrasound at the bottom while examining a segment of the CCA in the longitudinal view. UltraFast™ Doppler demonstrated excellent flow sensibility and provides a framerate of more than 80 Hz.

**Figure 4 diagnostics-12-01168-f004:**
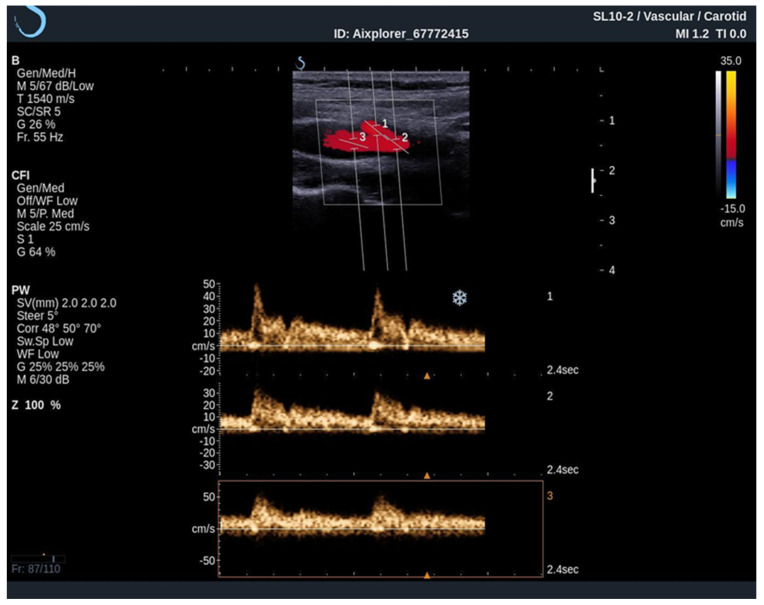
An UltraFast™ Doppler examination of the bifurcation of the carotid artery. Several measurements can be performed independently of each other with a high degree of reliability since the acquisition is made during the same cardiac cycle. In this example, spectra from the ICA (3) and ECA (2) are analyzed simultaneously, one can differentiate the ICA from the ECA on the basis of spectral morphology with ECA demonstrating a high-resistance spectrum and the ICA demonstrating a low-resistance spectrum.

## Data Availability

Not applicable.

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
