# Peer review of "Contribution of UltraFast™ Ultrasound and Shear Wave Elastography in the Imaging of Carotid Artery Disease"

_diagnostics, 2022, doi:10.3390/diagnostics12051168_

Round 1

Reviewer 1 Report

The article presents a compendium of the latest methods of non-invasive examination of carotid arteries. The study presents both the benefits and limitations of those methods in a critical and reliable manner. Figures are a valuable addition to the content of the article. I have no critical remarks and I propose to accept the article in present form.

Reviewer 2 Report

In this paper, Bulum et al reviewed the literature regarding carotid artery disease and its diagnostic techniques using their experiences. 
The picked topic is obviously crucial, and needs updates frequently. Please see my comments below; 
*Title could be more specific, there are many imaging techniques, and it is all about USG.
*Figures are not telling anything, I cannot understand without reading legends. It could be more explainable, like illustration, at least one of them.
*Some grammar and typo errors must be corrected.
*You should put your references at the end of the sentences you cited, at least at the end of the paragraph.
*You do not need to talk about CV disease in the conclusion part. You should be more specific, and give focused conclusions.
